# Uncovering financial distress conditions and its determinant factors on insurance companies in Ethiopia

**Samuel Godadaw Ayinaddis**[1]*, **Habtamu Getachew Tegegne**[2]

**1** Department of Management, Woldia University, Woldia, Ethiopia, **2** Department of Accounting and Finance, Woldia University, Woldia, Ethiopia

\* samuel.g@wldu.edu.et

**Data Availability Statement:** "All relevant data are within the paper and its Supporting Information files."

**Funding:** The author(s) received no specific funding for this work.

## Abstract

The study examines the financial distress situation and its determinants in insurance sectors in Ethiopia. To achieve study objectives, revised Altman's 2000 is adopted to measure the financial distress situation. The study adopted an explanatory research design with an arrangement of secondary data analysis via document analysis, quantitative approach, and deductive method of inquiry. The study used panel data from ten insurance companies *over* the study period 2010/11-2020/21. Descriptive and regression analyses were performed to analyze the data using STATA 14. Econometric model estimation procedures and multiple regression assumptions were tested accordingly. The random effect regression result revealed that firm-specific factors (liquidity and profitability) have a significant positive association, whereas firm size significantly negatively impacts financial distress. While the random effect regression result also proposed inflation has a positive and significant association with financial distress. However, firm-specific factors (revenue growth and leverage) have positive and negative, respectively, and macroeconomic factors (Gross Domestic Product) have positive but statically insignificant to the financial distress situation of insurance sectors in Ethiopia.

## Introduction

Financial distress is "the likelihood of bankruptcy which depends on the level of liquid asset as well as on credit availability" [1]. It is a circumstance where a firm cannot pay its obligation [2, 3]. This shows that the firm is in a position of minimum cash flow [4]. The cash flow is not adequate to satisfy current obligations [5], Which upshots their failure [6] and bankruptcy [3], according to The Traditional Gambler ruin theory developed by Feller 1968, where a gambler wins or loses money by chance. The author represents the "P" likelihood of win and (1-P) possibility of loss. This game continues until the gambler is out off [7]. The firm can continue as gamblers playing with some possibility of loss, continuing to operate up to its net worth reached zero [8]. The major drawback of this theory is that it does not condider the existence of security market access [9].

**Competing interests:** The authors have declared that no competing interests exist.

**Abbreviations:** CPI, Consumer Price Index; EBIT, Earnings Before Interest and Tax; GDP, Gross Domestic Product; ROA, Return on Asset; ROE, Return on Equity.

However, the static *Trade-Off Theory*, which was formulated by [10], states that the use of debt raises the value of the firm; for the persistence of financial distress, the company's necessary understand the optimum capital mix selection within the firm [11]. There is a certain level at which further use of debt becomes unfavorable, and continuous use of debt will maximize agency and bankruptcy costs which have the consequence of reducing the organization's worth leading to the possibility of financial distress. However, many academicians need to clarify this issue in establishing the optimum capital structure and an appropriate mixture of debt and equity [11]. According to [12], this theory is not a suitable predictor since the theory tells about only changes in capital structure without providing further information.

In addition, the *Picking Order Theory*, initially suggested in 1984 by Donaldson and modified via [13], states that firms can follow a particular source of finance. First, they use internally generated cash as the principal source of finance, and if the firms do not have enough internally generated cash, it resorts to debt financing. Finally, equity funding is the last resort of financing. The critical suggestion of this theory is that an increase in the use of external sources of funds (debt and equity) may affect the firms negatively; this aggravates the probability of financial distress [14].

Furthermore, according to cash management theory, the case of financial distress and business failure is the existence of improper cash management of the firm (Aziz and Dar, 2006). The firms should manage cash inflow and outflow to avoid fund imbalance. During the cash period, outflow surpasses flow since dividend payment and tax build-up (Kosikoh, 2014). On the other period, cash inflow exceeds outflow, and sales and borrowers may realize quickly (Kosikoh, 2014). The persistence of cash imbalance may cause financial distress to the organization and lead to business failure [8, 15]. Therefore, management has to pay much attention to the cash management of the firm to protect the firm from financial distress [16].

According to the entropy theory, the fundamental mechanism for identifying a firm's financial distress could be a careful look at the change in the statement of financial position [8]. This theory employs univariate (used accounting base ratio and market indicator and multivariate analysis (more than one variable is analyzed at a time) [17] to examine the structure of the balance sheet [18]. However, if the firm cannot maintain an equilibrium state of asset and liability and cannot control it in the future, it is more likely to fall into distress [8].

Financial distress is the stage of deterioration in the financial condition of firms and occurs before liquidation [19, 20]. The latest global financial crisis (2007–08) affects many international firms and institutions [21]. According to the World Banks 2010 report as cited in [22], in Europe, the United States, and Asian countries financial distress has become a primary concern for the financial sector worldwide [21].

In the current competitive and globalized business environment, the fittest firm will survive, and other firms that face financial distress will terminate [23]. Generally, financial distress not only distracts the firm's financial system but also impairs its organizational structure [18]. Recently, the issue of financial dusters of firms has become one of the hot and debated, and controversial issues in the field of finance due to the collapse of giant companies in developing and developed nations in the 2008 world economic crisis and the adverse effect on the corporate firm [24, 25]. Ethiopia was no exception to this phenomenon and engaged in financial difficulties due to the global financial crisis in 2008. Raw material price reduction in 2009 led to Ethiopian currency devaluation [26], and distress tartrates the organization's survival [25].

Therefore, the renewed interest among scholars, practitioners, and academicians in investigating determinants of financial distress at the firm level is accelerated. As a result, predicting financial distress has recently gained countless attention for researchers [27]. Different empirical reviews suggested that firms' financial distress determinants are numerous and vary across countries and regions [28]. According to [29], a common cause of financial distress and failure

is a complicated mix of different problems and symptoms. Giant and profitable firms engaged themselves in trouble due to rapid expansion and the introduction of tough competitors (Zwaig and Pickett, 2012) as cited in [9]. Young companies have also struggled with financial distress due to capital inadequacy [30].

According to [31, 32], both internal and external factors contribute to financial distress. It usually occurs due to the presence of weakness either directly or indirectly by managers' decision-making and failure in the strategy set of the firm [31]. Whereas, [15] states that family-owned company's less likely to fail in financial distress. [33] also suggested that the critical cause of financial distress for firms is endogenous variables compared to exogenous variables. However, a study by [34] states that macroeconomics has a significant effect on financial distress. Whereas, [35] concludes that macroeconomic factors do not significantly affect financial distress.

In Ethiopia, a study carried out by [26] concluded that firm-specific factors (firm age, liquidity, operational viability, and accurate corporate governance) have a positive and significant impact on the financial distress conditions of selected beverage and manufacturing firms. [36] suggested that liquidity risk and net income growth are positive and significant, and solvency risk has a negative and significant contribution to the financial distress condition of commercial banks in Ethiopia.

Furthermore, [37] suggested that bank-specific factors (firm size, leverage, profitability, and firm age) strongly affect financial distress. In contrast, the tangibility of assets has a significant positive impact on the financial distress proxy measured by Altman's Z score in Ethiopian insurance sectors. Finally, the researcher concludes that insurance sectors are in the safe zone. Likewise, a study conducted by [38] states that the financial health condition of insurance companies in Ethiopia is not a safe zone.

Therefore, this study was demanded in Ethiopia because different theoretical views and empirical studies contradicted each other. From the empirical studies' point of view, as per the researcher's knowledge, in Ethiopia, few studies were conducted on the determinants of financial distress in the insurance sectors. While previous studies conducted in Ethiopia also did not consider macroeconomic (external) factors. Hence, this study examines financial distress condition and its determinant factor on insurance companies in Ethiopia measured by Altman's Z score by considering both firm-specific (firm size, leverage, liquidity, revenue growth, and profitability) and macroeconomic (inflation and GDP) factors.

## Objective and contribution of the study

Overall, the study makes a significant contribution to the understanding of financial distress conditions and its firm-specific and macroeconomic factors determinants in the insurance industry in Ethiopia. The findings of the study can be used by insurance companies, policy-makers, regulators, and other stakeholders to make informed decisions and take appropriate actions to improve the financial health of the insurance industry in Ethiopia.

The main objective of this study was to examine financial distress conditions and their determinant factor on insurance companies in Ethiopia with the following specific objectives.

✓ To measure the extent of financial distress in insurance sectors in Ethiopia

✓ To examine the effect of firm-specific factors (firm size, leverage, liquidity, revenue growth, and profitability) on insurance sectors in Ethiopia

✓ To examine the effect of macro-economic factors (inflation and gross domestic product) on insurance sectors in Ethiopia

### The hypothesis of the study

**H01** Firm-specific factors (firm size, leverage, liquidity, revenue growth, and profitability) do not have a relationship with firms in financial distress

**H02** Macro-economic factors (inflation and gross domestic product) do not have a relationship with firms in financial distress

## Literature review and theoretical framework

In this section, the researcher identified a sample of previous studies regarding the role of financial distress conditions and its determinants from different countries, such as Malaysia, the USA, Kenya, Indonesia, and Ethiopia.

Another study by [15] examines the factors influencing financial distress among listed and OTC firms in Taiwan's emerging market. Using the Cox regression model, the research finds that liquidity, profitability, capital structure, and corporate governance significantly affect the likelihood of bankruptcy/delisting or recovery. Lower cash holdings, fewer independent directors, smaller control rights deviation, non-family-owned businesses, and high debt levels increase the probability of financial distress. Conversely, higher outsider shareholdings or control rights deviation decrease the likelihood of bankruptcy/delisting or increase the probability of recovery. Excess cash does not necessarily aid in resuming operations. Recovered firms outperform the market index, and industry classification and family-owned business status have minimal influence. The period from financial distress to bankruptcy/delisting or recovery averages 18 and 23 months, respectively.

An empirical study conducted by [39] in Malaysia on factors affecting the financial distress of publicly listed companies, the study used 101 companies from Bursa Malaysia during the period 2005–09 using long-term debt to total equity ratio and short-term debt to total equity ratio as the dependent variable. The study suggested that profitability; liquidity, growth and risk, solvency, and firm size positively and significantly impact financial distress. In contrast, the growth of operating profit has negative relation. Likewise, a study conducted by [40] from 1992–02 on ownership structure and the likelihood of financial distress in the Netherlands. The study states that firms with a higher level of managerial shareholding experience less financial distress and large outside shareholders reduce the possibility of financial distress. Finally, the study concluded that a high level of institutional shareholding with a low likelihood of financial distress on Dutch firms on the Amsterdam stock exchange.

Additionally, a study conducted by [41] on financial and non-financial factors to financial distress in listed insurance sectors in Indonesia states that cash flow. Liquidity, independent commissioner, and ownership do not contribute to financial distress. On the other hand, the study conducted by [42] on the financial health condition of retail companies in the Indonesia stock exchange from 2012–17. The study concluded that retail companies in telecommunication devices, building materials, convenience store models, and minimarkets had more potential to experience financial distress. Whereas the debt-to-equity ratio and the interest rate hurt distress, gross demotic product positively affects financial distress.

Further, in a study conducted by [4] on determinants of financial distress in selected 15 insurance companies in Kenya, solvency margin and net debt. The researcher concludes that profitability, liquidity, efficiency, and leverage significantly impact the firm's financial distress. Firm size had a significant moderating effect on distress. Finally, the researcher concludes that insurance sectors in Kenya face financial distress measured by the Altman Z score value. Whereas a study conducted by [33] in Kenya from 2009–12 on the case of financial distress in industrial and commercial development corporations. The researcher concluded that the leading financial distress case is endogenous variables rather than exogenous ones. Further, the

study notes that inadequate capital, improper capital allocation, access to credit, poor accounting record, poor internal management, and shortage of skilled workforce are the leading case of financial distress of firms.

Furthermore, a study conducted by [43] the study suggested financial ratios (liquidity ratio, leverage ratio, and profitability ratio). Profitability has a negative and significant effect; liquidity has no effect. Leverage has a positive and significant effect in predicting the financial distress of small and medium scale enterprises in Malang city. The study concludes that for medium-sized companies, it may be used as a consideration for taking corrective action before it evolves into severe financial distress and leads to bankruptcy.

While the study conducted in Ethiopia by [44] through evaluating the financial distress condition of a microfinance institution in Ethiopia (2010–2015), the researcher used revised Altman's revised Z- score. The study concludes that 94% of microfinance institutions are in the safe zone. Another study conducted in Ethiopia's banking sector from 2002–12 on six private banks suggested that capital to loan ratio and net interest income to total revenue ratio have a positive and significant impact on financial distress measured by Altman's Z score [45]. However, non-performing loans significantly negatively impact the financial health of selected private banks.

Finally, the study conducted by [37] on financial distress and its determinant in selected insurance sectors in Ethiopia from 2008–19. The study used firm-specific factors (firm size, profitability, leverage, and company age) that were negatively correlated with financial distress. On the other hand, asset tangibility and loss ratio positively and significantly impact Likewise, in a study by [25] from 2010–19 on determinants of financial distress in Amhara region manufacturing share companies in Ethiopia, the researcher measures financial distress by debt service coverage ratio. The study suggested that firm-specific factors (liquidity, profitability, solvability, asset size) and macroeconomic factor inflation has positive and statistically significant effect on the debt service coverage ratio. At the same time, the gross domestic product has a positive but insignificant effect on the financial distress of manufacturing firms in the Amhara region.

## Conceptual framework of the study

The internal and external factors (independent variables) that determine firm's financial distress condition is presented (Fig 1). The conceptual framework of the study is presented as follows.

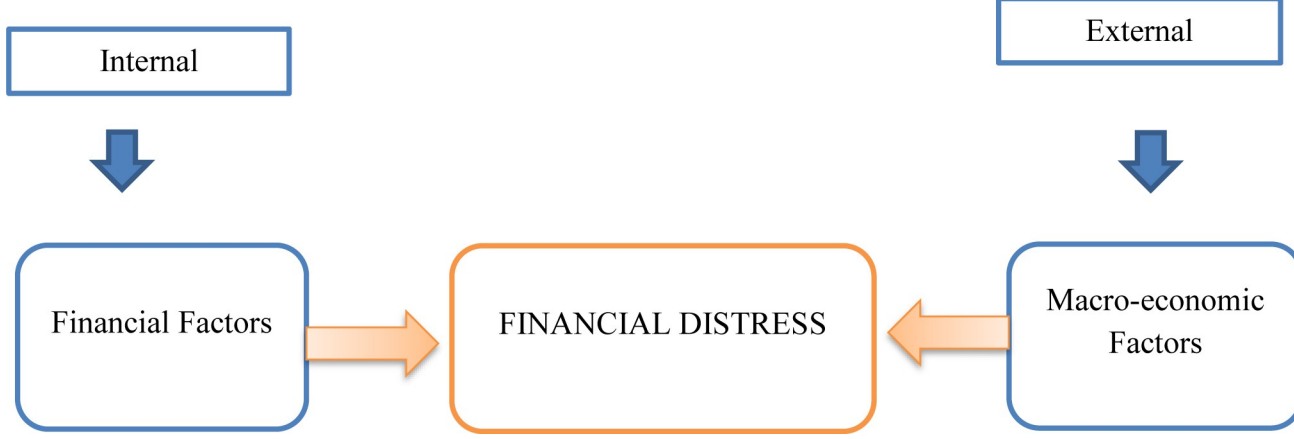

**Fig 1. Conceptual framework on determinants of financial distress.** Source: Own development.

## Research methodology

**Research design and research approach.** In this study, the researcher used an explanatory research design. The dimension of this design is an arrangement of secondary data collection via ex-post, longitudinal time dimension. However, the study explained the financial distress condition and its determinants of selected insurance companies in Ethiopia. Additionally, the study employed a quantitative research approach and a deductive method of inquiry. The rationality behind this quantitative approach is relevant due to its analysis of numerical data by applying statistical tests (inferential and descriptive statistics) [46]. Furthermore, the study used a deductive method of inquiry.

**Data source and method of data collection.** To achieve the objective of the study, the researcher used secondary data from the study period 2010–2020. Data for this study were obtained from annual financial statements of the insurance companies obtained from the National Bank of Ethiopia (NBE) and World Bank. Currently, 17 recognized insurance companies are operating in Ethiopia. In this study, the researcher selected 10 insurance companies' available audited financial statements for ten consecutive years from 2013 to 2022 were selected purposefully.

**Model specification.** To examine the financial distress condition of selected insurance sectors, the researcher used the revised original Z score model $Z''$, which was designed for both private, public, and manufacturing and non-manufacturing firms [47]. (Eq 1).

$$Z'' = 6.56X1 + 3.26X2 + 6.72X3 + 1.05X4 \qquad \text{Eq 1}$$

$$X1 = \frac{\text{Working Capital}}{\text{Total Asset}}$$

$$X2 = \frac{\text{Retained Earning}}{\text{Total Asset}}$$

$$X2 = \frac{\text{Earnings before Interest and Tax}}{\text{Total Asset}}$$

$$X2 = \frac{\text{Book Value of Equity}}{\text{Total Liability}}$$

$$Z'' = \text{Overall Index}$$

If the $Z''$ value is >2.60, they are classified as non-bankrupt firms; if their index value ranges between 1.10 to 2.60, they are classified as in the gray area, and if index values are less than 1.10, companies are in a difficult situation and classified in the high risk of bankruptcy. The accuracy of this model was 97% for non-bankrupt firms and 90.9% for bankrupt firms [19, 47] model was used by researchers like [45, 48]. Further, according to [49], the discriminant analysis given by Altman is adequate and accurate in predicting financial distress.

This study employed OLS Panel data multiple regressions since panel data offer a quick and affordable way to monitor how people's opinions and behaviors change over time, and it can improve model parameter inference accuracy while simplifying computation and statistical inference [50, 51]. The following regression models were used by the researcher with some modifications depending on prior studies on the issue under investigation [14, 38, 44, 45, 52].

**Table 1. Summary for measurement of dependent and independent variables.**

| Variables | | Variable Name | Measurement |
|---|---|---|---|
| **Dependent variable** | | **Financial distress** | **Revised Altman Z score** |
| Independent variables | Firm-specific factors | Firm size | Log of total asset |
| | | Liquidity | Current asset/Current liability |
| | | Leverage | Total debt/total equity |
| | | Profitability (ROA) | Net income /total asset |
| | | Revenue growth | $\frac{EBIT_t - EBIT_{t-1}}{EBIT_{t-1}}$ |
| | Macro-economic factors | Gross domestic product | Yearly growth domestic product |
| | | Inflation | Consumer price index |

The researcher used the following regressions equation to examine the effect of financial distress, macroeconomic and firm-specific factor on financial distress. (Eq 2).

$$FD_{it} = \beta_0 + \beta_1 LIQ_{it} + \beta_2 FS_{it} + \beta_3 Rg_{it} + \beta_4 lEV_{it} + \beta_5 PROF_{it} + \beta_6 INF_{it} + \beta_7 GDP_{it} + U_{it}$$

Eq 2

Where,
FD = dependent variable, which is the output of the Z" score
LIQ = Liquidity
FS = Firm size
Rg = Revenue growth
LG = Leverage
PROF = Return on asset
INF = Inflation
GDP = Gross domestic] roduct
$\beta_0$ = constant
$U_{it}$ = error component for bank "i" at time t, and it assumed to have zero mean $u_{it} = 0$
$\beta_1, 2, 3, 4, 5,$ and 6 are parameters to be estimated i = Insurance companies t = time periods

**Variables description and measurements.** In the following table, the study variables and their corresponding measurements are presented (see Table 1).

**Measurement of dependent variable: Financial distress.** Financial distress is when a firm has managerial, operational, and financial difficulties [30]. To measure a firm's financial distress, the researcher used the revised Altman, 2007 Z score. This method was used by different researchers (45, 48]. Accordingly, the financial distress of firms can be determined using the above equation on model specification.

## Measurement of an independent variable (firm-specific factors)

**Profitability.** Previous Studies used different measuring methods to measure a firm's profitability (financial performance). Those measuring methods are return on asset and return on equity. ROA is used as a measurement of financial performance. The reason for choosing ROA as the vital proxy for bank profitability instead of the alternative return on equity (ROE) is that an analysis of ROE disregards financial leverage and its risks [53]. ROA indicates how profitable a company is of its total assets. It gives an idea of how efficiently the management uses assets to generate revenue. ROA = Net income/Total assets [54].

**Firm size.** Bank size is considered an essential determinant of financial distress [14, 39]. According to [55], size has a negative contribution to growth. When an investor holds a large volume of capitalized stocks, indicates the economy's unfavorable state may be low, unstable,

and depressed. Furthermore, [56] suggested that small firms have a likelihood of failing than big firms because large firms have better market experience than small firms, with limited resources and finance. Previous researchers [37, 57] used a log of the total asset to measure bank size. In this study, the researcher used the logarithm of the total asset to measure bank size.

**Leverage.** It shows how heavily the firm is in debt. Financially distressed organizations suffer from huge debt burdens characterized by high-interest payments. Leverage (LEV) is the ratio of total debt to total assets [58]. It is the amount of debt a firm has in proportion to its equity capital. Leverage in this study was measured by the company's total debt / total asset value.

**Revenue growth.** Organizational growth is another factor determining financial distress [32, 39]. The growth of the firm is mainly measured by revenue growth. A decrease in revenue growth is an early warning signal of the financial distress of firms [32]. Earning growth can be determined by (EBITt −EBITt-1)/EBITt-1 [37].

## Measurement of an independent variable (macroeconomic factors)

**Gross domestic product (GDP).** Economic growth measured by gross domestic product shows the general economic condition of a given nation. Firms face financial distress when the economy worsens [25]. It is the most commonly used macroeconomic indicator, as it measures total economic activity within an economy [59]. In this study, GDP is measured using the yearly growth domestic product.

Inflation rate: loss of a currency's purchasing power results in a general and sustained price increase [25]. In this study, the consumer price index measured the inflation rate. Inflation; annual inflation rate of Ethiopia measured by the average consumer price index (CPI).

## Data presentation discussion and recommendations

**Financial distress situation of insurance sectors in Ethiopia.** To achieve the objective of the study, the researcher evaluated the financial health conditions of each insurance company in Ethiopia. Eleven years of financial data from the period (2010–2020) through using the revised Altman Z" score analysis model. As indicated in **Fig 2** [60]. Below is shown the financial distress condition of insurance companies in Ethiopia fluctuated from period to period.

On the other hand, when examining the average financial distress condition of insurance sectors in **Table 2** below (Global (2.38) and Lion (2.82) insurance companies) are in the gray zone since their value is less than 2.60. While the remaining eight insurance companies lie in the safe zone. Since their financial distress situation value measured by Altman is >2.60.

**Diagnostic test for classical linear regression model (CLRM) assumptions.** Before proceeding to the result of regression analysis, diagnostic tests were carried out accordingly to check whether the model's data fit the basic classical linear regression model assumptions. Such as the zero mean value of disturbance, normality test, multicollinearity test, Brusch-pagan test, and Wooldridge test were tested accordingly. All assumptions are satisfied in the regression equation. (See S1 Appendix) Furthermore, the researcher used the F-test, Breusch, pagan lagrange multiplier, and Hausman-test to choose the appropriate panel data estimation for this study. Finally, the best model used in this study was the random effect regression model (**Table 3**).

## Empirical results and discussions

**Leverage and financial distress.** The results of the random effect model regression revealed that leverage measured by total debt divided by total assets is a positive and

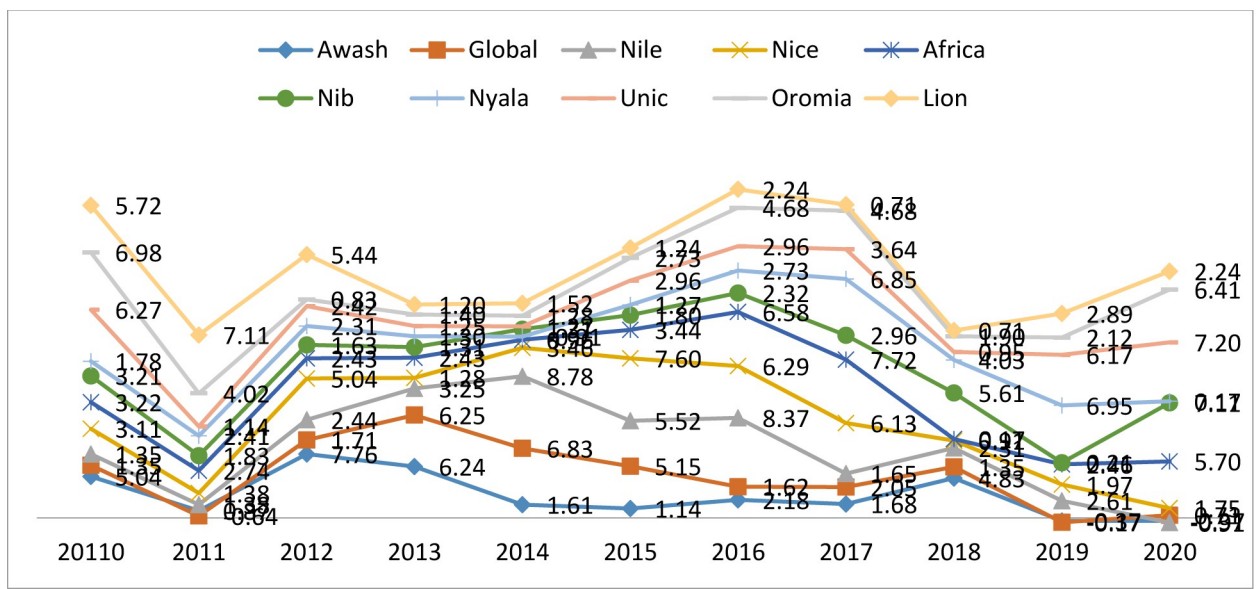

**Fig 2. Altman's Z" score values of each insurance company in Ethiopia from (2010–20).** Source: Own analysis (2023).

statistically insignificant relationship with the financial distress of insurance sectors in Ethiopia. The result of the study is consistent with [61], who states that liquidity has a significant positive impact on the financial distress of no financial firms. In addition to this [62], as cited in [14], concludes that leverage has a positive impact on financial distress [34, 63].

Contrary to this study, a study conducted by [14, 34, 63, 64] suggested that financial distress will increase when rising in firm leverage. Financially distressed organizations frequently suffer from vast debt burdens characterized by high-interest payments. While [25, 38, 57, 65, 66] rivaled that leverage has negative relation with financial distress.

**Profitability and financial distress.** The regression coefficient of profitability is positive and significant (**also in Table 3**). The result indicates that profitability measured by ROA is an essential determinant of the financial distress of firms; the result of this study is consistent with [38, 39, 41, 61, 67, 68]. An organization with low profitability leads to a low level of liquidity, which may affect firms in meeting obligations and, finally, firms exposed to financial distress [32].

Table 2. Financial distress situation of insurance sectors in Ethiopia measured by Altman's (2,000) from 2010–20.

| No | Insurance companies | Average distress situation measured by Altman's (2000) |
|---|---|---|
| 1 | Awash | 2.78 |
| 2 | Global | 2.38 |
| 3 | Nile | 3.34 |
| 4 | Nice | 3.54 |
| 5 | Africa | 3.44 |
| 6 | Nib | 2.66 |
| 7 | Nyala | 2.63 |
| 8 | Unic | 3.29 |
| 9 | Oromia | 3.37 |
| 10 | Lion | 2.82 |

Source: Own analysis (2023)

**Table 3. Random effect regression result.**

| Wald chi2(7) = 100.58 | | Prob>chi2 = 0.000 | | |
| --- | --- | --- | --- | --- |
| Number of obs. and groups | | [110,10] | | |
| R-sa; Within, between and overall | | [0.5348, 0.0562,0.4965] | | |
| Zz | Coef. | Std. Err. | Z | P>Z |
| LEV | .259844 | .232268 | 1.12 | 0.263 |
| ROA | 1.346086 | .2710774 | 4.97 | 0.000** |
| LIQ | .1087678 | .0458993 | 2.37 | 0.018* |
| Rg | -.0052632 | .0364979 | -0.14 | 0.885 |
| FS | -.4185698 | .101366 | -4.13 | 0.000** |
| INF | .0099793 | .0048506 | 2.06 | 0.040* |
| GDP | .1583435 | .1919869 | 0.82 | 0.410 |
| Cons | 1.895633 | 2.433579 | 0.78 | 0.436 |

Source: Own analysis (2023)

Note

** and * indicates at **5%** and **1%** level of significance

**Liquidity and financial distress.** The random effect model regression coefficient of liquidity positive and significant relation to financial distress; the result of the study is consistent [39, 57, 63, 65, 66], indicate that liquidity has a positive link with financial distress.

Contrary to this study, [39] showed that liquidity and financial distress have a negative relation. Similarly, [62, 64, 69] suggested that an increase in liquidity leads to the decreased financial distress of firms. Finally, [61] concluded that liquidity has no significant impact. In addition, a study by [70] concludes that firms with low liquidity probably experience financial due to the inability of such organizations to meet their recurring obligations.

**Firm size and financial distress.** The coefficient of firm size is negative and significant at 5%. This show suggests that larger insurance companies are more likely to expose financial distress. The result of the study is similar to [71–75], who suggested firm size is a significant variable influencing financial distress negatively. Further, [34] stressed that the likelihood of financial distress is expected to increase when firm size increase. Further, the result of this study supports firm size has a negative contribution to growth. When an investor holds a large volume of capitalized stocks, indicates the economy is unfavorable state may be low, unstable, and depressed.

On the contrary, this study [25, 39, 41, 66] suggested that firm size has a significant positive impact on financial distress. Additionally, [56] suggested that big firms have less probability of failing than small firms since big firms have market experience, unlimited connections, and unlimited financial resources. Small firms need better market experience, connections, and financial resources. On the other hand, a study conducted by [38, 57] concluded that firm size does not determine financial distress.

**Earning growth and financial distress.** The regression coefficient of earning growth is negative and insignificant. The result of the study is consistent with [39, 68]. Those researchers suggested that earnings growth measured by revenue growth is an essential determinant of financial distress; a decrease in revenue is an early warning signal of financial distress for firms.

Additionally, [6, 76–78] found a significant negative relationship between revenue growth and financial distress. On the other hand, this study's result contradicts [38, 39].

**Inflation and financial distress.** In this study, inflation has a positive and significant effect on financial distress. An increased inflation rate creates uncertainties within the business

environment and discourages investment. Also, inflation could make the export of firms less competitive in the global market and raise the cost of production of firms. It leads to a decline in firm profitability, and a persistent decline in firm profitability could lead to financial distress [32]. The result of the study is consistent with [62, 69, 79]. On the contrary, this study [34, 35] stated that inflation negatively determines financial distress.

**Economic growth and financial distress.** Economic growth measured by GDP significantly impacts the financial distress of insurance sectors in Ethiopia. The result is consistent with [25, 34], which suggested that economic growth has positive determinants of the financial distress of firms. On the other hand, the result of this study contradicts with other studies [35].

## Conclusion and recommendations

Based on the study's findings, the financial distress condition of insurance sectors measured by revised Altman's Z" score (2000) fluctuated from period to period. While the average financial distress situation for Global and Lion is in the gray zone, and the remaining eight insurance companies lie in the safe zone. Further, the random effect regression result shows that firm-specific factors (profitability measured by ROA and liquidity) and macro factors (inflation) have positive and significant determinants for distress. In contrast, firm size has negative and significant determinants of the financial distress situation of insurance sectors in Ethiopia.

Inflation has a positive determinant of financial distress conditions in selected insurance sectors. When the purchasing power of money is reduced, it creates uncertainty in the business sector, which makes it less competitive in the global market; consequently, firms engaged in financial distress. Additionally, Large insurance firms are more likely to expose financial distress, and the likelihood of financial distress is expected to increase when firm size increase. profitability is another crucial determinant of financial distress since the higher the level of profitability, the better financial healthiness and stability. Organizations with low profitability leads to low level of liquidity, which may affect firms in meeting obligations and, finally, firms exposed to financial distress.

**Finally,** the researcher forwarded the following recommendation based on the study's findings.

✓ Insurance companies should develop policies based on the appropriate level of significant variables to improve operations and reduce periodic fluctuations in financial distress.

✓ Financial managers and the board of directors needs to set up risk management and internal control measures to detect early warning signal of financial distress and to take early preventive measures.

✓ The board of directors and financial managers must pay decisive attention to firm-specific and external factors since the government's macroeconomic policies directly impact firms' operations.

✓ The government should design business-friendly policies to intensify growth and sustainability.

Overall, while there is no guaranteed way to eliminate financial distress, effective management involves identifying potential sources of distress and taking appropriate actions to mitigate the risk of default. In every situation, managers must carefully consider a variety of factors such as costs, timing, project status, operational hazards, and reputational damage, and make decisions accordingly. While the optimal solution may not always be apparent, managers need to act decisively and implement strategies tailored to the specific circumstances at hand. By doing so, they can minimize the risk of financial distress and ensure the long-term success of their organization.

### Ideas for future research

The present study addresses the financial distress conditions and its firm-specific and macro-economic factors determining insurance companies in Ethiopia. However, there are several suggestions for future research that could build on this finding. These include:

1. Conducting a comparative study: Future research could compare the financial distress conditions and their determinant factors in insurance companies in Ethiopia with those in other African countries or even globally. This would provide a broader perspective on the factors that affect the financial health of insurance companies.

2. Including qualitative data: While this study mainly relied on quantitative data, future research could incorporate qualitative data such as interviews with key stakeholders in the insurance industry. This would provide more insights into the factors that contribute to financial distress and how they can be addressed.

3. Investigating the role of regulatory policies: this study identified macroeconomic factors that contribute to financial distress in insurance companies. Future research could focus on examining the impact of regulatory policies such as capital requirements, solvency regulations, and other regulatory frameworks on the financial health of insurance companies in Ethiopia.

4. Including other variables: this study considers some firm-specific and macro-economic variables, while, future researchers could examine size, value, and business cycle variables.

### Supporting information

**S1 Appendix.**
(DOCX)

### Author Contributions

**Conceptualization:** Samuel Godadaw Ayinaddis.

**Data curation:** Habtamu Getachew Tegegne.

**Formal analysis:** Habtamu Getachew Tegegne.

**Investigation:** Habtamu Getachew Tegegne.

**Methodology:** Samuel Godadaw Ayinaddis.

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
