## [Decision Letter · Decision Letter 0]

10 Jul 2023

PONE-D-23-18855Uncovering Financial Distress Conditions and Its Determinant Factors On Insurance Companies in Ethiopia.PLOS ONE

Dear Dr. Ayinaddis,

Thank you for submitting your manuscript to PLOS ONE. After careful consideration, we feel that it has merit but does not fully meet PLOS ONE’s publication criteria as it currently stands. Therefore, we invite you to submit a revised version of the manuscript that addresses the points raised during the review process.

We look forward to receiving your revised manuscript.

Kind regards,

Ricky Chee Jiun Chia

Academic Editor

PLOS ONE

Journal Requirements:

Wang, Ma-Ju & Shiu, Heng-Ruei. (2014). Research on the common characteristics of firms in financial distress into bankruptcy or recovery. Investment Management and Financial Innovations. 11. 233-243. 

In your revision ensure you cite all your sources (including your own works), and quote or rephrase any duplicated text outside the methods section. Further consideration is dependent on these concerns being addressed.

   "I have no competing interests."

5. Please ensure that you include a title page within your main document. You should list all authors and all affiliations as per our author instructions and clearly indicate the corresponding author.

6. Please ensure that you refer to Figures 1 and 3 in your text as, if accepted, production will need this reference to link the reader to the figure.

Reviewers' comments:

Reviewer's Responses to Questions

**Comments to the Author**

1. Is the manuscript technically sound, and do the data support the conclusions?

Reviewer #1: Yes

2. Has the statistical analysis been performed appropriately and rigorously? 

Reviewer #1: I Don't Know

3. Have the authors made all data underlying the findings in their manuscript fully available?

Reviewer #1: Yes

4. Is the manuscript presented in an intelligible fashion and written in standard English?

Reviewer #1: No

5. Review Comments to the Author

Reviewer #1: The review report is attached in pdf.

Referee Report on

PONE-D-23-18855

“Uncovering Financial Distress Conditions and Its Determinant Factors On Insurance Companies in Ethiopia”

Submitted to “PLOS ONE”

July 05, 2023

I have carefully reviewed this interesting manuscript; However, I have numerous suggestions.

Suggestions and Comments

1. The current exposition of the manuscript is like a thesis, not a precisely written scientific study. In other words, I suggest reducing the length of the introduction and literature review sections.

2. After a long discussion on well-established theories and concepts in the introduction section, the authors suddenly summarized the first part of the introduction, saying, “Therefore, this study was demanded in Ethiopia because different theoretical views and empirical studies contradicted each other. Previous studies conducted in Ethiopia also did not consider macroeconomic (external) factors. Hence, this study examines financial distress condition and its determinant factor on insurance companies in Ethiopia measured by Altman's Z score”. Comment: The earlier comments did not focus on any concept/need that emphasizes on studying financial distress in “Ethiopia”. Please improve the flow of the study.

3. In “Statement of the Problem”, the authors stated that “Over and done with considering macroeconomic (economic growth and inflation) and firm-specific factors (firm size, leverage, liquidity, revenue growth and profitability), whereas those researchers in Ethiopia overlooked macroeconomic factors”. Comment: Economic growth and inflation are not the Macroeconomic variables? Please specify this statement, such as “other macroeconomic variables”, or rewrite this statement.

4. As stated above, inflation and economic growth have been discussed in the context of Ethiopia by earlier studies, then what is the contribution of this study? Hypotheses 6 and 7 are the only hypotheses discussing Macroeconomic variables, and they are the same inflation and GDP (GDP=economic growth).

5. A supporting study that provides empirical and theoretical evidence on the link between business cycle variables and other factors, such as size or B/M variables, can be discussed: DOI: https://doi.org/10.3390/economies6010014. It may help authors to connect different points.

6. The way Fig. 3 is presented is not a scientific way. It should be tabulated. The snapshot of Stata output as a graph can only be in an online appendix or supplementary materials. Revise it, please.

7. The data section mentions 17 insurance firms, whereas Table 2 and Figure 2 show 10 firms. … ???

8. Why Panel data multiple regression method is opted? Provide literature support and also define regression models employed: GLS/OLS/CLMR and so on, or whatever it is.

9. The four points discussed before the subsection, "Ideas for Future Research, " are common sense. It is common knowledge to consider (i) the appropriate level of significant variables, (ii) risk management and internal control measures to use, (iii) pay attention to firm-specific and external factors, and (iv) government should design a favorable strategy. Please emphasize some creative recommendations.

Overall, the presentation of the (i) manuscript, (ii) first 3 sections and subsections, and (iii) tables/figures is poor. The authors should restructure the manuscript and extend constructive discussion on the study's main results and methods. After these suggestions are incorporated, the manuscript can be considered for publication.

6. PLOS authors have the option to publish the peer review history of their article (what does this mean?). If published, this will include your full peer review and any attached files.

Reviewer #1: **Yes: **Fahad Ali

---

## [Author Response · Author response to Decision Letter 0]

16 Sep 2023

Subject: Revision and Resubmission of PONE-D-23-18855

Uncovering Financial Distress Conditions and Its Determinant Factors On Insurance Companies in Ethiopia.

PLOS ONE

Dear Dr. Ricky Chee Jiun Chia,

Thank you for giving us the opportunity to submit a revised draft of my manuscript titled Uncovering Financial Distress Conditions and Its Determinant Factors On Insurance Companies in Ethiopia” for publication in PLOS ONE. We appreciate you and all anonymous reviewers for your precious time in reviewing this manuscript and providing valuable comments. We have carefully included the reviewer comments and revised the manuscript accordingly to reflect the suggestions provided. We have highlighted modifications in yellow in the manuscript file attached. The authors welcome any further comments and concerns. Below is a point-by-point response to the reviewers’ comments and concerns.

With warm regards, 

Samuel Godadaw Ayinaddis, On behalf of the co-author

samuel.g@wldu.edu.et, 

Editor’s and Reviewer’s Comments

Author Response: We thank you so much. We have updated the revised manuscript based on the journal’s requirement for format and style.

Wang, Ma-Ju & Shiu, Heng-Ruei. (2014). Research on the common characteristics of firms in financial distress into bankruptcy or recovery. Investment Management and Financial Innovations. 11. 233-243. 

In your revision ensure you cite all your sources (including your own works), and quote or rephrase any duplicated text outside the methods section. Further consideration is dependent on these concerns being addressed.

Author Response: We appreciate the reviewer’s assessment. We have cited and included the aforementioned reference in the revised manuscript.

 "I have no competing interests."

Author Response: We appreciate the reviewer’s assessment. We have revised it accordingly.

Author Response: Thank you for the comment. We have attached the output that supports our study as a supplementary file for the reader’s reference. However, we are unable to attach all the raw data used in our study. This is because we purchased the data from the National Bank of Ethiopia and it cannot be accessed publicly.

5. Please ensure that you include a title page within your main document. You should list all authors and all affiliations as per our author instructions and clearly indicate the corresponding author.

Author Response: The title page has been included in the main document which includes all authors and their affiliation, email address and the corresponding author indication.

6. Please ensure that you refer to Figures 1 and 3 in your text as, if accepted, production will need this reference to link the reader to the figure.

Author Response: We have revised the citation of the figures and tables in the text. Thank you for the suggestions.

Reviewers' comments:

Comments to the Author

1. Is the manuscript technically sound, and do the data support the conclusions?

Reviewer #1: Yes

Author Response: Thank you.

2. Has the statistical analysis been performed appropriately and rigorously?

Reviewer #1: I Don't Know

Author Response: We ask apologies for any confusion from the reviewer. However, the research has been done in an appropriate and rigorous way following all the necessary statistical procedures. 

3. Have the authors made all data underlying the findings in their manuscript fully available?

Reviewer #1: Yes

Author Response: Thank you.

4. Is the manuscript presented in an intelligible fashion and written in standard English?

Reviewer #1: No

Author Response: Thank you for your feedback on our article. We appreciate your suggestions and have carefully considered them in our revision. We have revised the language of the article to improve clarity and readability of the English. In addition, we have engaged a professional editor to ensure that the manuscript is free from errors and consistent in style throughout. We believe that these changes have significantly improved the quality of the article and addressed the concerns you raised. We hope that you find our revised manuscript satisfactory and look forward to hearing from you soon.

5. Review Comments to the Author

Reviewer #1: The review report is attached in pdf.

I have carefully reviewed this interesting manuscript; However, I have numerous suggestions.

Author Response: Thank you for the acknowledgement.

Reviewer’s Comments

1. The current exposition of the manuscript is like a thesis, not a precisely written scientific study. In other words, I suggest reducing the length of the introduction and literature review sections.

Author Response. We appreciate the reviewer’s assessment. Based on the comments, the introduction and statement of the problem is merged and re arranged and the number of pages reduced to 3.5 pages (1,334 words) from 4.25 pages (1,508 words). Also, the literature review section has been updated accordingly. 

2. After a long discussion on well-established theories and concepts in the introduction section, the authors suddenly summarized the first part of the introduction, saying, “Therefore, this study was demanded in Ethiopia because different theoretical views and empirical studies contradicted each other. Previous studies conducted in Ethiopia also did not consider macroeconomic (external) factors. Hence, this study examines financial distress condition and its determinant factor on insurance companies in Ethiopia measured by Altman's Z score”. Comment: The earlier comments did not focus on any concept/need that emphasizes on studying financial distress in “Ethiopia”. Please improve the flow of the study.

Author Response. Thank you. We carefully examined the introduction and statement of the problem an updated based on the reviewer’s suggestion. We believe the flow of the study improved. 

3. In “Statement of the Problem”, the authors stated that “Over and done with considering macroeconomic (economic growth and inflation) and firm-specific factors (firm size, leverage, liquidity, revenue growth and profitability), whereas those researchers in Ethiopia overlooked macroeconomic factors”. Comment: Economic growth and inflation are not the Macroeconomic variables? Please specify this statement, such as “other macroeconomic variables”, or rewrite this statement.

Author Response: Thank you very much. It was a typo error. We have revised it now. The revised paragraph is read as follows: 

“Previous studies conducted in Ethiopia also did not consider macroeconomic (external) factors. Hence, this study examines financial distress condition and its determinant factor on insurance companies in Ethiopia measured by Altman's Z score through considering both firm specific (firm size, leverage, liquidity, revenue growth and profitability) and other macro-economic (inflation and GDP) factors”.

4. As stated above, inflation and economic growth have been discussed in the context of Ethiopia by earlier studies, then what is the contribution of this study? Hypotheses 6 and 7 are the only hypotheses discussing Macroeconomic variables, and they are the same inflation and GDP (GDP=economic growth).

Author Response: We have revised it based on the suggestion. As per the researcher knowledge previous study does not consider macro-economic variables in insurance sector. That is why this study examine through considering macro-economic variable page 

5. A supporting study that provides empirical and theoretical evidence on the link between business cycle variables and other factors, such as size or B/M variables, can be discussed: DOI: https://doi.org/10.3390/economies6010014. It may help authors to connect different points.

Author Response: Thank you for your recommendation. We carefully examine the study and cite the study appropriately and used to strengthen our study (page and ).

6. The way Fig. 3 is presented is not a scientific way. It should be tabulated. The snapshot of Stata output as a graph can only be in an online appendix or supplementary materials. Revise it, please.

Author Response: Thank you for the assessment. We revised it and convert into table.

 7. The data section mentions 17 insurance firms, whereas Table 2 and Figure 2 show 10 firms. … ???

Author Response: Sorry for the confusion we caused. The number 17 firms indicate the total population (licensed insurance organization in Ethiopia) while, we selected 10 insurance sectors as a sample for the study.

 8. Why Panel data multiple regression method is opted? Provide literature support and also define regression models employed: GLS/OLS/CLMR and so on, or whatever it is.

Author Response: well appreciate the recommendations. we carefully examined and add the following statement through supporting 2 key literature Andre, H. J. (2017) and Hsiao, C. (2007.

9. The four points discussed before the subsection, "Ideas for Future Research, " are common sense. It is common knowledge to consider (i) the appropriate level of significant variables, (ii) risk management and internal control measures to use, (iii) pay attention to firm-specific and external factors, and (iv) government should design a favorable strategy. Please emphasize some creative recommendations.

Author Response: after examine our previous recommendations we believe we add new recommendations for managers before sub section and further researchers in the sub section.

---

## [Editor Report · Decision Letter 1]

3 Oct 2023

Uncovering Financial Distress Conditions and Its Determinant Factors On Insurance Companies in Ethiopia.

PONE-D-23-18855R1

Dear Dr. Samuel Godadaw Ayinaddis,

We’re pleased to inform you that your manuscript has been judged scientifically suitable for publication and will be formally accepted for publication once it meets all outstanding technical requirements.

Kind regards,

Ricky Chee Jiun Chia

Academic Editor

PLOS ONE
---

## [Editor Report · Acceptance letter]

10 Oct 2023

PONE-D-23-18855R1 

Uncovering Financial Distress Conditions and Its Determinant Factors On Insurance Companies in Ethiopia. 

Dear Dr. Ayinaddis:

I'm pleased to inform you that your manuscript has been deemed suitable for publication in PLOS ONE. Congratulations! Your manuscript is now with our production department. 

Kind regards, 

on behalf of

Dr. Ricky Chee Jiun Chia 

Academic Editor

PLOS ONE